# Cloud-Based Remote Sensing for Wetland Monitoring—A Review

Abdallah Yussuf Ali Abdelmajeed *, Mar Albert-Saiz, Anshu Rastogi and Radosław Juszczak

Laboratory of Bioclimatology, Department of Ecology and Environmental Protection, Faculty of Environmental Engineering and Mechanical Engineering, Poznan University of Life Sciences, Piątkowska 94, 60-649 Poznań, Poland

\* Correspondence: abdallah.abdelmajeed@up.poznan.pl

**Abstract:** The rapid expansion of remote sensing provides recent and developed advances in monitoring wetlands. Integrating cloud computing with these techniques has been identified as an effective tool, especially for dealing with heterogeneous datasets. In this study, we conducted a systematic literature review (SLR) to determine the current state-of-the-art knowledge for integrating remote sensing and cloud computing in the monitoring of wetlands. The results of this SLR revealed that platform-as-a-service was the only cloud computing service model implemented in practice for wetland monitoring. Remote sensing applications for wetland monitoring included prediction, time series analysis, mapping, classification, and change detection. Only 51% of the reviewed literature, focused on the regional scale, used satellite data. Additionally, the SLR found that current cloud computing and remote sensing technologies are not integrated enough to benefit from their potential in wetland monitoring. Despite these gaps, the analysis revealed that economic benefits could be achieved by implementing cloud computing and remote sensing for wetland monitoring. To address these gaps and pave the way for further research, we propose integrating cloud computing and remote sensing technologies with the Internet of Things (IoT) to monitor wetlands effectively.

**Keywords:** wetland; peatland; cloud computing; remote sensing monitoring; PaaS; SaaS





## 1. Introduction

Wetlands are transitional environments where terrestrial and aquatic zones exist and some properties of the two zones are shared. Thus, they have an extremely crucial role in the hydrological cycle. Moreover, wetlands have numerous similarities but differ in size, location, and hydrology. They occupy around 10% of the global land surface area and have been ranked as some of the most diverse ecosystems on Earth [1]. Additionally, they are characterized by different water depths, hydric soils, and wetland-adapted flora and fauna [2,3]. Therefore, in terms of definition and classification, wetlands have been defined and classified differently by different nations and organizations, depending on various perspectives, e.g., country policy, wetland application, vegetation types, or water level [2,4–7]. One example of the classification is provided by the Canadian Wetland Classification System [8], which identifies five types of wetlands:

1. Bog—an ombrotrophic peatland dominated by sphagnum moss species;
2. Fen—a minerotrophic peatland dominated by graminoid species and brown mosses;
3. Swamp—a peatland or mineral wetland dominated by woody vegetation;
4. Marsh—a minerotrophic wetland with periodic standing water or slow-moving water, dominated by graminoids, shrubs, forbs, and emergent plants;
5. Shallow water—a minerotrophic wetland where water is up to 2 m deep for most of the year and has less than 25% of emergent or woody plants.

Wetlands are included among the world's most productive ecosystems. They provide several eco-services, such as water purification, removal of pollutants, carbon regulation, protection from natural threats, soil and water conservation, enhanced biodiversity, diverse wildlife habitats, recreational activities, fish and shellfish aquaculture, and flood

mitigation [5–7]. These ecosystems are also home to one-third of at-risk vegetation and animals [3,9].

Furthermore, from an economic perspective, wetlands are highly significant due to the wide-ranging applications they provide to populations through the economy and their support to the sustainability and pliability of communities [10]. These applications incorporate provisioning administrations (food, fresh water, and fuel), managing administrations (climatic guidelines, hydrological guidelines, contamination control, disintegration security, and relief of normal risks), social administrations (profound, instructive, and strict), and supporting administrations (biodiversity, soil arrangement, and supplement cycling). In addition, wetlands provide environmental benefits and are esteemed more profoundly than many other biological systems on Earth [10]. Nevertheless, notwithstanding wetlands' manifold advantages, wetland conversion and depletion have surpassed 50% worldwide [11] and reached up to 87% since the onset of the 18th century [11,12]. Furthermore, anthropogenic activities such as drainage, groundwater extraction, intense irrigation practices, and urban and agricultural land replacement have degraded wetlands globally [13].

Natural events such as climate change and natural catastrophes (i.e., wildfire, flood, drought) have also contributed to wetland degradation [4]. Furthermore, global and local wetlands have a substantial cycle (seasonal and annual). Due to disturbance, they become inconsistent in their spatial extent, which is subject to variations in water balance components (evapotranspiration, precipitation, and runoff) that result in fluctuations of the water table and lead to changes in vegetation composition [14]. Therefore, monitoring wetlands is crucial for obtaining precise, consistent, and up-to-date information about the attributes of wetlands, such as the extent of change, type, and status.

Remote sensing (RS), in combination with wetland science, is currently better used than in recent times for accurately quantifying wetland status and changes over time. Wetland mapping utilizing earth observation data is critical for regional, national, and global natural resource management, as RS technology captures data about the Earth's surface ranging from low to ultra-high resolutions. However, due to wetlands' varied and fragmented environments and the spectral similarities of different wetland types [15,16], reliable wetland monitoring is complex, especially at a broad scale. Globally, precise, consistent, and complete national- or provincial-scale wetland inventories are still insufficient, with most research focusing on producing local-scale maps using limited remote sensing data [17]. Therefore, one of the ways to effectively address the issues mentioned above is to use cloud computing (CC) technology.

According to the National Institute of Standards and Technology (NIST), "Cloud computing is a model for enabling convenient, on-demand network access to a shared pool of configurable computing resources (e.g., networks, servers, storage, applications, and services) that can be rapidly provisioned and released with minimal management effort or service provider interaction" [18]. CC solutions offer extremely dependable data centre design, including load balancing, real-time backup, remote disaster recovery, big data service distributed via the internet, and computing service from anywhere [19]. CC has been widely applied in RS for several years [20]. It is worth mentioning that, regarding the efficiency and analysis process, CC delivers efficient and systematically arranged information which directly impacts the final beneficiary of the process, as compared to conventional computing. In this case, it is conducive for the beneficiaries (e.g., researchers, scientists, and governmental organizations) to monitor the wetlands' changes and processes, providing instantaneous decision support.

CC is primarily defined based on two different aspects: the service model and the deployment model, which can be private, hybrid, or public depending on the level of privacy it supports. The CC model consists of the three most common service models: platform as a service (PaaS), software as a service (SaaS), and infrastructure as a service (IaaS). Apart from the three service models, there are some models associated with big data, such as data storage as a service (DaaS) and function as a service (FaaS) [21].

There are existing applications of cloud-based RS, such as Google Earth Engine (GEE), Microsoft Planetary Computer, and Earth on Amazon Web Services (AWS). Cloud computing reduces the efforts spent in in situ measurements and processing time. Moreover, it reduces human error while measuring and obtaining raw data, especially in wetlands (due to their heterogeneity).

The main aims of this paper are to form state-of-the-art cloud computing applications in wetland monitoring, highlight the importance of cloud computing for RS data, and show the benefits of utilizing such a technology in wetland monitoring. The subsequent research questions were formulated to accomplish these objectives:

1. Which cloud computing service model has been utilized in wetland monitoring?
2. How widely utilized are the different monitoring applications of remote sensing data on wetlands using cloud computing, and what are their limitations and accuracy?
3. Which monitoring strategies were performed using cloud computing technology?
4. What economic gains can be realized from integrating cloud computing and remote sensing data in the monitoring of wetlands?

This systematic literature review (SLR) is organized into several key sections. Section 2 outlines the methodology used to conduct the SLR and provides the conceptual framework. Section 3 presents the results in two subsections: the first describes the general statistics related to the included studies, and the second answers the research questions and discusses the findings. Section 4 identifies the limitations and potential threats to the review. Sections 5 and 6 present the conclusion and future work, respectively.

## 2. Methodology

The present study adhered to a systematic methodology describing the framework and structure of the research: (1) The initial step involved defining the principal aim of the review clearly and concisely, reflecting the scope and purpose of the investigation; (2) Forming the research questions to achieve the main aim; (3) Selecting the databases for searching the literature; (4) Identifying the workflow for the article selection process, shown in Figure 1. Since steps 1 and 2 were mentioned in the introduction, step (3) was executed by selecting two databases, Scopus and Web of Science (WoS), where the publications were searched. We used the following search string to narrow down the literature on the searched databases, as shown in Table 1.

To be included in the study at least one of the terms "CLOUD COMPUTING" OR "google earth engine" OR "MAAP" OR "Multi-Mission Algorithm and Analysis Platform (MAAP)" OR "Giovanni" OR "NASA POWER" OR "earth data" OR "Nebula" OR "Copernicus" and one of the terms "Wetland" OR "peatland" OR "bog" OR "fen" OR "swamp" OR "mire" OR "marsh" had to appear in either the title, abstract, or keywords. Furthermore, we included only peer-reviewed research articles on cloud computing applications and remote sensing in the monitoring of wetlands published in English with no time constraints. The process of retrieving relevant articles was performed in June 2022, thus encompassing all published works on the topic until that time, as included in this review. Out of 265 articles retrieved from the database search, 172 met the established criteria after removing duplicate articles, and were considered for manual screening.

The last step of the screening phase was completed manually for the selected literature records based on reading the full text. The articles had to meet the following selection criteria: (1) Articles which utilized only remote sensing without cloud computing were excluded; (2) Only full-text, original, peer-reviewed articles presenting in-depth experimental results in English language were included; (3) The articles had to utilize both cloud computing and remote sensing to be included; (4) Some of the studies displayed the terms of search queries in the abstract and title for examples such as "FEN" and "Giovanni", but they did not investigate the wetlands or any of the other related features; thus, such articles were excluded. After all these steps, 50 full-text articles (see Supplementary File for details) were identified and analysed based on the following extracted data:

A.    The number of annual published articles;

B.    The distribution of studies per country;
C.    Utilized cloud computing platforms and remote sensing data;
D.    Observed temporal coverage of time series analyses;
E.    Frequencies of the spatial scales studied;
F.    Frequencies of remote sensing platforms;
G.    Frequencies of methods applied.

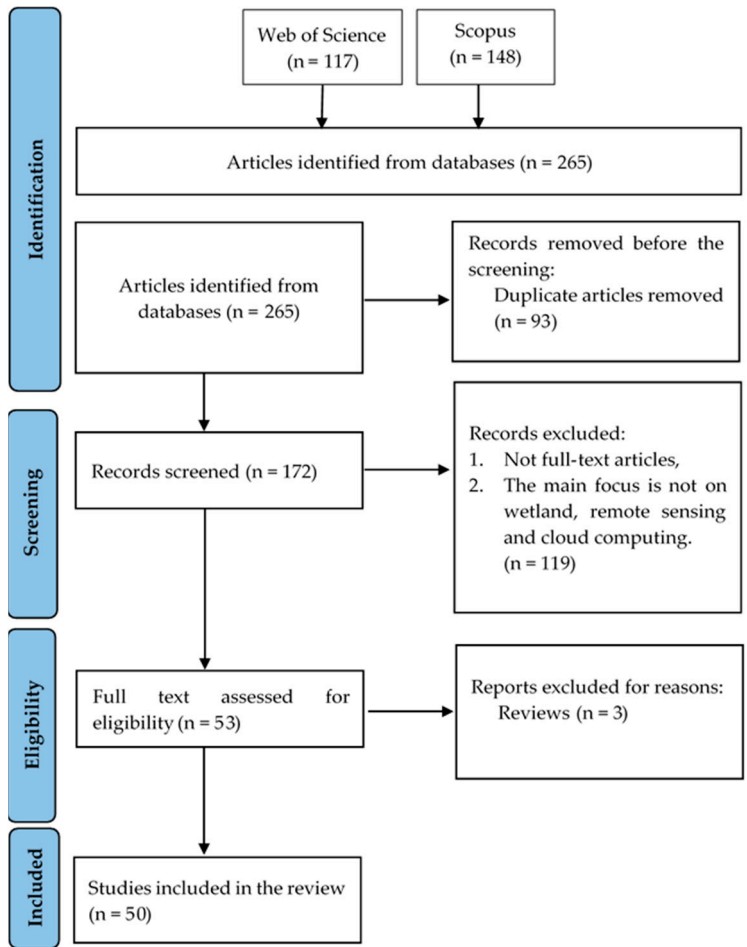

**Figure 1.** Systematic literature review, schematic representation, and reasons for exclusion.

**Table 1.** Used search queries for searched databases.

| Database | Search Query |
| --- | --- |
| Scopus | TITLE-ABS-KEY ("Cloud Computing" OR "Google Earth Engine" OR "MAAP" OR "Multi-Mission Algorithm and Analysis Platform (MAAP)" OR "Giovanni" OR "NASA POWER" OR "earth data" OR "Nebula" OR "Copernicus") AND TITLE-ABS-KEY ("Wetland" OR "peatland" OR "bog" OR "fen" OR "swamp" OR "mire" OR "marsh") AND (LIMIT-TO (LANGUAGE, "English")) AND (LIMIT-TO (DOCTYPE, "ar") OR LIMIT-TO (DOCTYPE, "re")) |
| Web of Science | TS = ("Cloud Computing" OR "Google Earth Engine" OR "MAAP" OR "Multi-Mission Algorithm and Analysis Platform (MAAP)" OR "Giovanni" OR "NASA POWER" OR "earth data" OR "Nebula" OR "Copernicus") AND TS = ("Wetland" OR "peatland" OR "bog" OR "fen" OR "swamp" OR "mire" OR "marsh") |

## 3. Results

This section presents key findings and responses to the research questions. Moreover, it shows the general statistics related to the primary studies selected for this systematic literature review (SLR).

*3.1. General Statistics*

3.1.1. Annually Published Papers

Fifty primary studies met the eligibility criteria; thus, they were selected and included in this review, as described in the supplementary material (Table S1). Figure 2 shows the distribution of these studies by year of publication. The topics related to the application of cloud computing and remote sensing in wetlands started in 2016, and their popularity has increased in recent years: around 58% of articles (29 papers) were published in just one and half years (from January 2021 till June 2022). The findings above illustrate the emerging trend among researchers dealing with wetland monitoring to employ cloud computing (CC) technology. As such, it is reasonable to anticipate a surge in research to develop remote sensing CC-based applications for wetland monitoring soon.

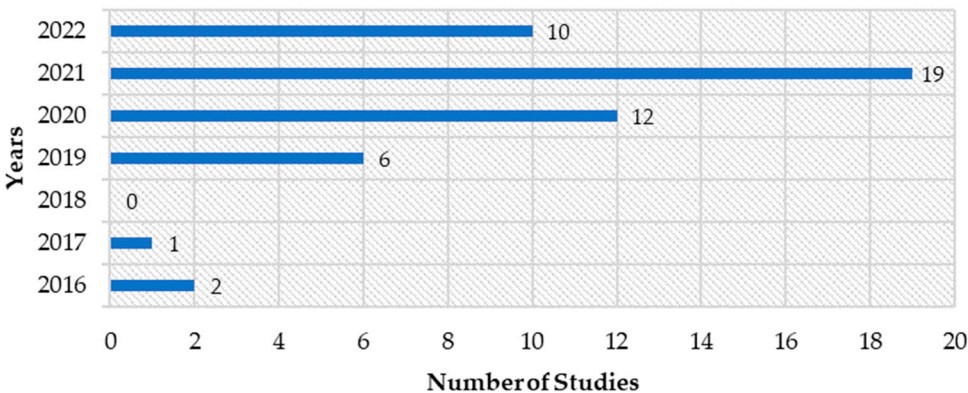

**Figure 2.** The annual number of selected primary studies on remote sensing and cloud computing-based studies of wetland monitoring.

3.1.2. Retrieved Primary Study Classification Based on the Journal in Which the Articles Were Published

The studies included in this SLR were published in twenty-three journals from eight well-known publishers (Figure 3). First, most of the articles were published by *MDPI* (46%), then by *Elsevier* (18%), *Springer* (12%), *Taylor & Francis* (8%), *IEEE* (6%), *IOP* (4%), and *PLoS* (4%), followed by the *American Society of Agricultural and Biological Engineers* (*ASABE*, 2%), arranged from the highest to the lowest contribution, respectively.

3.1.3. The Spatial Distribution of Studied Wetlands in the Selected Articles

Twenty-four percent of the articles were related to Chinese wetlands, while Canadian and wetlands in the USA were represented by 22% and 18% of articles, respectively (Figure 4). Countries such as Iran, Turkey, and Costa Rica had at least two studies associated with each, whereas the rest contributed with a single study. It is noteworthy that several studies were conducted in multiple countries, including Gxokwe et al.'s investigation [22] spanning South Africa, Botswana, Mozambique, and Zimbabwe; Hardy et al.'s study [23] conducted in various southern African nations such as Barotseland, Zambia, and the Zambezi Region; and a further examination by Zhang et al. [24] which was implemented in China, South Korea, and North Korea (Figure 4).

Although wetlands exist in most countries worldwide, in every climatic zone, from the polar regions to the tropics and from high altitudes to dry regions, only CC and RS-related studies on wetlands in 30 countries were reported in the selected literature. It is worth mentioning that only three studies were placed in Europe, despite the significant number of wetlands on this continent [1,12].

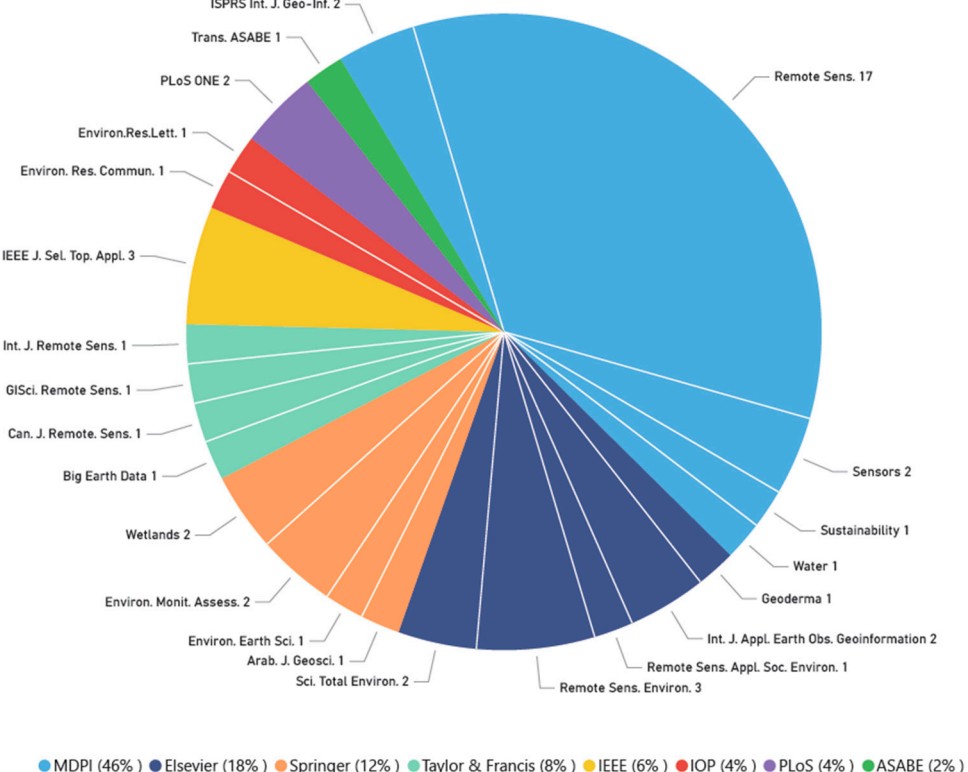

**Figure 3.** Publisher and journal contributions to the studies included in this SLR.

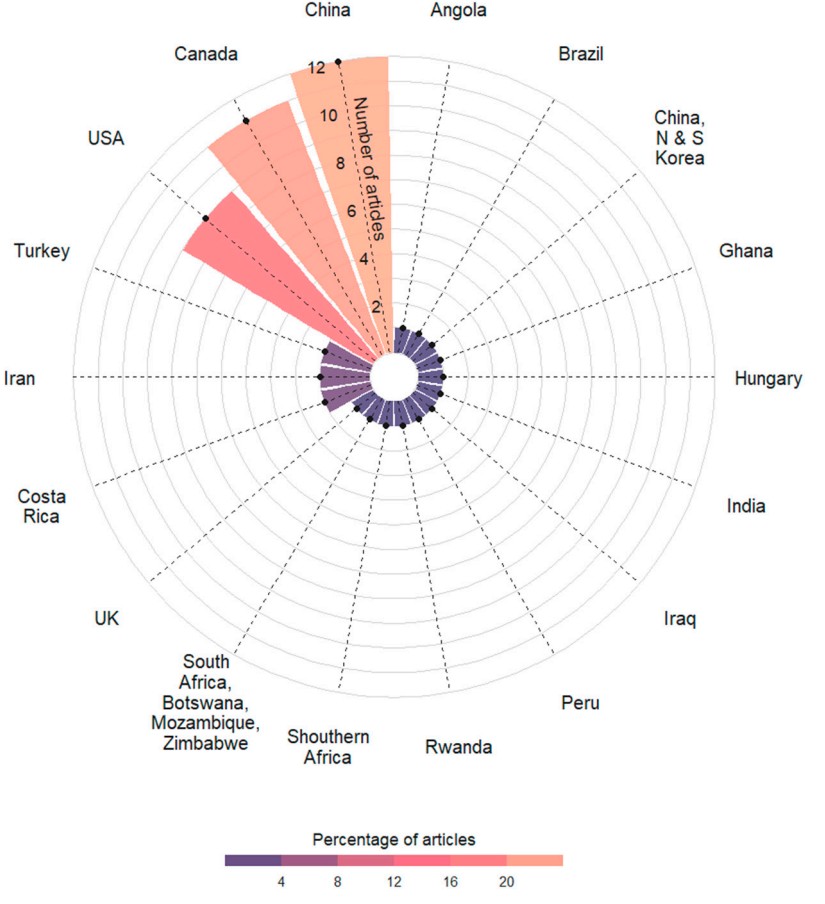

**Figure 4.** Number and percentage of publications per country.

3.1.4. Distribution of Selected Articles Based on First Authors

Canadian authors exhibited the highest published works among the included studies (Figure 5). Specifically, Mahdianpari M. and Amani M. contributed three articles each, while DeLancey ER contributed two. This value is relatively substantial compared to authors from other nations, such as China and the United States, where different first authors were credited for each published article.

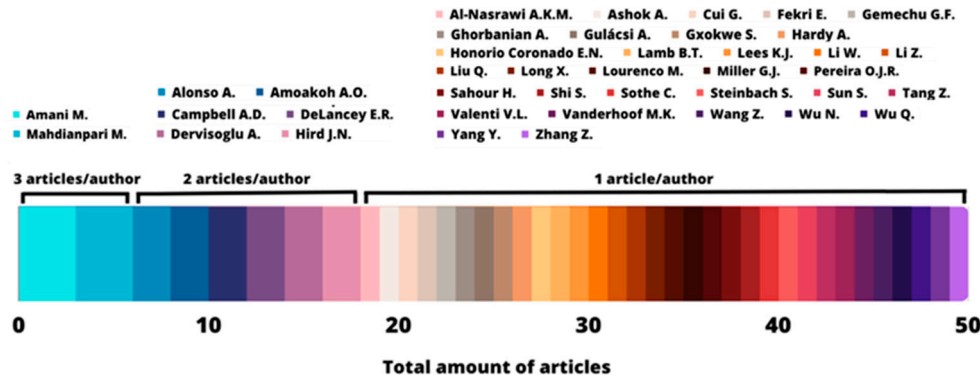

**Figure 5.** The author's contribution to the selected articles.

*3.2. Which Cloud Computing Service Model Has Been Utilized in Wetland Monitoring?*

All the papers included in this review used PaaS as a cloud computing model. PaaS is a development platform allowing researchers to develop and build RS-based applications directly on the PaaS cloud. Thus, SaaS and PaaS differ in that SaaS only hosts cloud applications that have already been developed, whereas PaaS offers a development platform that hosts both cloud applications that have already been developed and those that are still in development. PaaS must have a development infrastructure, including a programming environment, tools, configuration management, and other components, and an environment supporting application hosting to achieve this.

An example of PaaS is Google Earth Engine (GEE), which is most commonly used to implement the study and achieve the goal of CC [22,25–32]. GEE is a freely available platform with open-sourced scripting frequently used to automate the RS data for the area of interest, mainly at a big scale. It has a vast dataset for the world, mainly from satellites with a variety of spatial resolutions, from coarse (1 km, MODIS) to high (10 m, Sentinel 2), which are good enough for monitoring wetlands at different scales, from the local and regional to the continental and global scales. At a local scale, which is often the wetland level where some research infrastructure and experimental sites with small plots are located, wetlands can be monitored either by application of CC on the SaaS model (time and cost-efficient) or by the traditional methods, i.e., carrying out the ground-based measurements using the instruments in the field (time and not cost-efficient). However, using dedicated CC-based software, data from the fixed sensors in the study area can be read and analysed, and then quantitative and visualized results can be shown in real time. These time and cost-savings help researchers and professionals closely track the wetland status and minimize potential risks (e.g., fire) and maintenance costs. PaaS and RS can integrate a vast quantity of data, tools, and programs; then, by linking many wetlands with the same technology, global monitoring and tracking of the changes of those study areas will be possible and straightforward, thanks to Digital Twin, IoT, and CC.

*3.3. How Widely Utilized Are the Different Monitoring Applications of Remote Sensing Data on Wetlands Using Cloud Computing, and What Are Their Limitations and Accuracy?*

The requirements of the data needed to monitor an ecosystem depend on the purpose of its monitoring. Five types of monitoring purposes have been selected as subclasses of wetland monitoring in the articles analysed (Table 2).

**Table 2.** The subclasses of wetland monitoring strategies in the analysed articles.

| Monitoring Strategies | References |
|---|---|
| Prediction (1 article) | [33] |
| Time series analysis (6 articles) | [28,30,34–37] |
| Mapping (11 articles) | [23,24,31,35,38–44] |
| Classification (15 articles) | [9,17,22,27,45–55] |
| Change detection (17 articles) | [25,26,29,56–69] |

In most of the studies, satellite data is the most used; it has been used alone or in combination with other types of remote sensing data (airborne, UAV, or in situ), independent of the monitoring strategy applied (Figure 6). The satellite data's widespread use is thanks to the open-access products of space agencies such as the European Space Agency (ESA) and the National Aeronautics and Space Administration (NASA), providing products with spatial resolutions of 10 m (Sentinel 2) or temporal coverages since 1972 (Landsat 1). These products were mostly used for the detection of changes in wetlands (34% of articles), wetland classification (30% of articles), and wetland mapping (22% of articles).

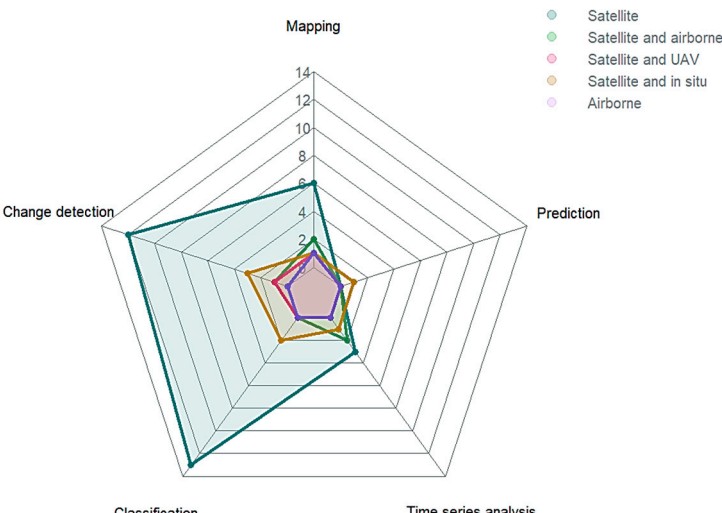

**Figure 6.** Distribution of the number of articles (y-axis) per monitoring strategy for each type of remote sensing data.

Only one article focused on predicting changes in wetlands, where a combination of in situ measurements and satellite data were applied to monitor plant phenology. However, weather conditions such as fog, made it impossible to use historical data to predict seasonality [33]. Additionally, climate change, with the variation in weather and seasonality, makes it more difficult to use historical data to predict future plant phenology as the future climate is unknown and unpredictable in the long term. Thus, approximately 12% of the articles primarily analysed time series data derived from satellite products. Three of them indicated problems with the number of images available [36], their spatial resolution [30], or the high level of moisture affecting vegetation index performance [37], but the average accuracy surpassed 90%. The accuracy refers from this point to the values provided by the authors in each publication, where accuracy is calculated from the percentage of hints compared to an already published result in the form of Corine land cover or ground control points grouped and averaged. The satellite products' low number of images and low spatial resolution was also a problem faced in combining satellite and airborne data [34]. Differences among wetland sites were identified as the next source of uncertainty when multiple watersheds or wetlands are studied combined and treated equally, where the addition of in situ measurements is recommended to avoid errors [28]. However, in situ data in combination with low spatial resolution satellite products has reported the problem

of pixels being too big [36], indicating that the combination with in situ data may still require high-resolution imagery either airborne or satellite. Furthermore, the accuracy was not reported in the studies combining airborne with satellite, and performing time series analysis [23,25,27]; hence, this SLR could not evaluate the accuracy.

In the case of wetland mapping, satellite data cannot provide enough information to differentiate among wetland classes, as it occurred for bogs and fens [42], and wetland types [32,44]. Another problem derived from satellite spatial resolution for mapping is the inability to distinguish water bodies, often masked by highly dense vegetation [23,43], or the inability to differentiate small fires; they are accounted for as giant fires, clustering several of them instead of treating them as small individual ones when burn severity is studied [39]. Five out of seven studies using satellite data for wetland mapping specifically reported problems with the spatial resolution of the imagery. In the only article using satellite data in combination with in situ measurements, a problem was reported with skewed data: a non-representative set of ground-based values was overestimating high values of soil organic carbon, for example, and underestimating low ones [41]. With the use of UAV data, the spatial resolution increased; however, when combined with satellite, the results faced issues with noise-free images to monitor phenology [24]. When combined with airborne data, multi-source satellite data were still needed for mapping due to the structural heterogeneity of some wetlands, such as peatlands [40].

Additionally, the low availability of airborne images per year could not capture rapid changes, such as inundation status, to map these processes [31]. Using satellite data with a higher resolution to validate results [38] can result in overestimating the actual accuracy (93.2% accuracy estimated), as it should be obtained from the comparison with ground-based data and not another remote sensing dataset. High-resolution data from UAVs or airborne missions have time, scale, and sometimes even price limitations, so their use also faces challenges [61]. Satellite data used for change detection analysis on wetlands faced the same problems with the heterogeneity and density of wetland vegetation, i.e., mapping, although the average accuracy of the results is higher (Table 3). However, the result can be biased because of the difference in the amount of articles for each monitoring type (Figure 6). The difference in spatial distribution and the spectral similarities among types of wetlands also caused challenges in change detection studies as it was impossible to select a specific shape and spectral indices to perfectly extract the changes over time [62,64,66]. The same can be observed when combining airborne and satellite datasets, [29,63] and satellite and in situ [65]. In contrast, for wetland mapping, the main challenge faced within change detection studies on wetlands was the processes of rapid changes caused by extreme events and strong seasonality of wetlands, with satellites not providing a high enough temporal resolution to monitor them [56,61,67]; the same occurred when combining it with airborne data [59]. The harmonic time series analysis can be used as a solution [68], increasing the accuracy from the average of 89% to 93.35% or by combining with UAV imagery, with 92% accuracy [69]. It is worth noting that the utilization of higher resolution satellite images for validating the results can potentially lead to an overestimation of accuracy. This is exemplified by the inundation and detection disturbances in land cover and their study of associated changes, which achieved a validation accuracy of 91.1% when using imagery from a private satellite, as opposed to in situ data [57].

Wetland classification was the second most used monitoring approach, with 15 articles, and was only surpassed by change detection (Figure 6). While not all articles in the remaining sections provide estimates of result accuracy due to insufficient ground-truth measurements, it is noteworthy that all classification analyses carried out in these studies include such estimates. The spectral similarities among wetland types or types of vegetation were again a source of errors (accuracies: 86–96%) [27,46]. However, the heterogeneity and complexity of the wetland ecosystem represented a more significant source of errors (with lower accuracies: 77–88%) [9,17], even with water bodies masked by dense vegetation [53]. The use of harmonic models can decrease the effect of these errors with accuracies of up to 91% [49]. High accuracies have been reported as average, though; sometimes, this is due to

the use of other satellite-based classifications to validate the result instead of in situ data [47]. A lack of ground-truth data can affect the validation of the study, decreasing the overall accuracy (72%) [54]. Similarly, training data are needed when applying deep learning techniques. If the training data are not enough, the classification out of the algorithm shows a lowered accuracy (68.7–77.1%) [22,45]. Using a satellite topography data approach to classify wetlands dropped the accuracy of a land cover classification (81.85%) due to the heterogeneity and complexity of wetlands [52]. However, some benefits are added when simultaneously merging synthetic aperture radar (SAR) and optical remote sensing data (82.7% accuracy), as the number of variables used for classification increased [50]. The addition of multiple satellite sources (>4), including private ones with high resolution, especially in combination with ground-truth data, allows wetland classifications with the high level of resolution needed (e.g., for studying floristic composition). These analyses perform with very high accuracy (96%), and an increase in the number of in situ variables is suggested to approach even higher accuracy [51]. The combination of ground-truth data and satellite images, including in situ data, can reach up to 97% accuracy in classifying vegetation types on wetlands, despite some confusion between wetland and non-wetland vegetation [48].

**Table 3.** Average accuracy per monitoring strategy for each type of remote sensing data. Note: airborne data were excluded, as only one article used this data type, and the accuracy was not reported.

| Monitoring Strategy | Satellite | Satellite + Airborne | Satellite + In Situ | Satellite + UAV | Total |
|---|---|---|---|---|---|
| Prediction | NA * | NA * | 67% | NA * | 67% |
| Time series analysis | 94% | NA * | 85% | NA * | 91% |
| Mapping | 82% | 94% | 83% | 94% | 86% |
| Classification | 84% | NA* | 97% | NA * | 85% |
| Change detection | 89% | 86% | 89% | 92% | 89% |
| Average total | 85% | 91% | 87% | 93% | 86% |

\* NA= not available.

The same sources of uncertainty are usually present independently of the type of monitoring used. The only exception appears with temporal-coverage-related uncertainties present for those using time series analysis, predictions, and detection of wetland changes. In most cases, the overall accuracy has been observed to increase over time due to enhancements in the quality of satellite data. For instance, in a study of land cover changes in a Chinese swamp, estimation accuracies improved from 82% in 1984 to 92% in 2018 [60]. However, no remote sensing data source can perform perfectly by itself. Additionally, applying robust algorithms in large areas implies a large use of computational resources [55]. Consequently, to better assess the uncertainties and improve the monitoring performance, a combination of automatic in situ meteorological stations and satellite and airborne/UAV data, validated with enough reliable in situ measurements using cloud computing services such as GEE, is recommended.

*3.4. Which Monitoring Strategies Were Performed Using Cloud Computing Technology?*

When deciding on the type of results, research can be classified according to two approaches: holistic and atomistic. The scale in which the area is analysed determines whether a generalized perspective is used (holistic), or a specific perspective is applied (atomistic). The number of details inside an area or the area covered is often prioritized. Thus, the type of wetland studied will vary with the scale. Minor scales usually focus on a particular type of wetland (e.g., marsh, bog, fen) as the limited area will include only that specific environment. With an increase in scale, the probability of identifying one type of wetland inside the area decreases, and the focus of the study will probably switch from specific vegetation identification to wetland delineation (Figure 7). From the articles reviewed, these two main approaches can be distinguished:

1.     Larger areas with a regional or national scale, including more than one type of wetland;

2. Smaller areas focused on a specific protected area with no more than two types of wetlands.

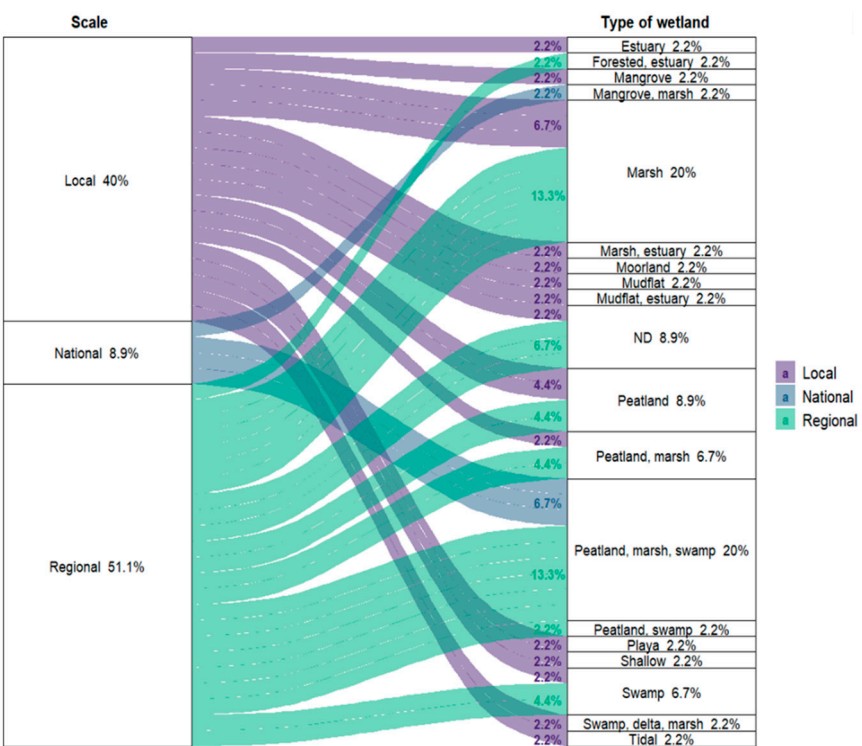

**Figure 7.** Percentage of papers depending on the scale used and type of wetland plus their distribution based on the relationship between these two parameters.

The type of results varies depending on ecosystem characteristics, as the type of vegetation [25,49,56], water surface changes [58,67], or even burn severity [39] are most often limited to local scale studies. Wetland delineation [63,64], distinguishing types of wetland [9,29,42,45,55], land use [65], or carbon content in the peat [41] are usually completed for larger scale studies. The performance of cloud computing has been suitable in each of the distinct scale–wetland combinations, although the challenges of using remote sensing and cloud computing vary. At the regional scale, the main challenges reported were the similarities between wetland types due to their spectral similarities and spatial heterogeneity. The spectral similarities produced higher confusion among wetland types, with an accuracy for bogs of 86%; reduced to 80% for fens [64]; and 80% in saline marshes studied in China [65]. In a study in the Great Lakes, these spectral similarities induced confusion among wetlands and uplands [27]. Small and highly vegetated wetlands as potholes masked water bodies [43]. Spectral similarities did not allow the differentiation of herbaceous vegetation [17]. The spatial heterogeneity of peatland constrained the results, not allowing the proper distinction of bogs and fens in Canada and reducing the accuracy to 69% [42]. Due to spectral similarities and spatial heterogeneity, the accuracy was reduced to 77% in a Canadian wetland inventory map [9].

Furthermore, the difference in shapes and distribution made monitoring alpine wetlands and swamps in the same study complex [62]. The same occurred with bogs, fens, swamps, and marshes on the island of Newfoundland [29], although bogs showed the highest producer accuracies between 92% and 97%, and fens had the highest user accuracies between 66% and 86%. At the national scale, the challenges were the lack of adequate data with high resolution [45] and low noise [24], or both [30], and the high computational resources for such large datasets [55]. The local-scale studies also faced challenges due to spectral and spatial features. The use of NDVI in wetlands can be compromised by the high moisture content, making it challenging to acquire the best results [37]. Problems

with the moisture or water table level were commonly faced at this scale [52] or due to its strong seasonality and fast changes [35,67]. The high heterogeneity of wetlands and spectral similarities, together with the moisture level, makes it almost mandatory to use multi-source approaches at local scales [40]. Apart from this, the same spectral similarities that do not allow the fine distinction among wetland types at a regional scale could make the reproducibility of algorithms hard when they are based on a particular type. Moreover, although high accuracy (96.44%) was displayed at a specific local scale, there is a large possibility that it will be restricted to a particular type of wetland [46]. The confusion occurs due to several uplands and the small areas covered by peatland, making monitoring difficult at the local scale [44,66].

For the excellent performance of the CC models, the generalization of scale adequacy for the CC methodology and its accuracy need to be evaluated. The decreased complexity in the analyses and the lower number of studies at the national scale resulted in the lowest accuracy among all the scales (Table 4). The second lowest was at the local scale, but a more comprehensive range of values for the standard deviation is shown in Table 4. Different monitoring techniques and distinct types of wetlands can be found inside each scale, and usually, the highest deviation is shown inside the groups with the most significant number of papers. This is why almost no deviation occurs at the national scale, as only four papers were found in this group. On average, individual (no more than two types) and mixed type of peatland studies show similar accuracies, 86.0% $\pm$ 41.0% and 86.9% $\pm$ 22.0%, respectively, but with a different number of papers analysed for each category (33 articles focused on individual types of wetlands and 18 mixed). As previously noted, the scale of wetland studies is often determined by the type of wetland under investigation, whether individual or mixed.

**Table 4.** Accuracy distribution depends on the study's scale, with the average represented as the average value $\pm$ standard deviation.

|  | National | Regional | Local |
|---|---|---|---|
| Average | 81.8 $\pm$ 9.6% | 86.9 $\pm$ 30.3% | 87 $\pm$ 42.5% |
| Max | 94% | 98.2% | 98% |
| Min | 71% | 69% | 67% |

Consequently, authors choose their study targets based on the specific wetland type, and their selection depends on the most suitable location for their research objectives. For this reason, accuracies are similar despite the number of types of wetlands studied. However, an accuracy increase will be expected when only one type of wetland and different performances depending on the type of wetland (with higher standard deviation as the results) are studied.

Random forest (RF) has been the most widely used when considering the machine learning method (ML) applied, with 19 articles applying it [9,17,24,27,28,32,40,44,45,47–51,55,56,61,64,65] (Figure 8). Ordered from the most to the least repeated, the other ML techniques used have been classification/regression trees (accuracy = 79.2%, [35,42,52,60]), clustering (accuracy = 98%, [31,66,67]), support vector machines (SVM, accuracy = 93.4%, [30,63]), and artificial neuronal networks (ANN, accuracy = 96.4%, [46]). However, when the unique method used was not RF, the authors preferred a mix of multiple ML techniques with lower accuracies than RF reported as the average of all methods applied (accuracy = 85.9%, [22,25,29,41,43,54,68,69]). On the other hand, not all the authors considered ML the technique needed, and index thresholding [35,37,40], object-based image segmentation [38], trend analysis [37,59], or regressions [33,53,57,62] have been successfully used for peatland monitoring (accuracy = 88.8%, [23,26,58]). Because not all papers have reported accuracies, and the number of papers between classes is not comparable, the analysis of the success of each technique cannot be assessed. For example, not using ML presents almost the same average accuracy as RF because only six articles from the thirteen included in this group reported

this value, while all the authors provided accuracy when using RF. ANN and SVM methods present the highest accuracies; this is not surprising considering that only three articles used these techniques [30,46,63].

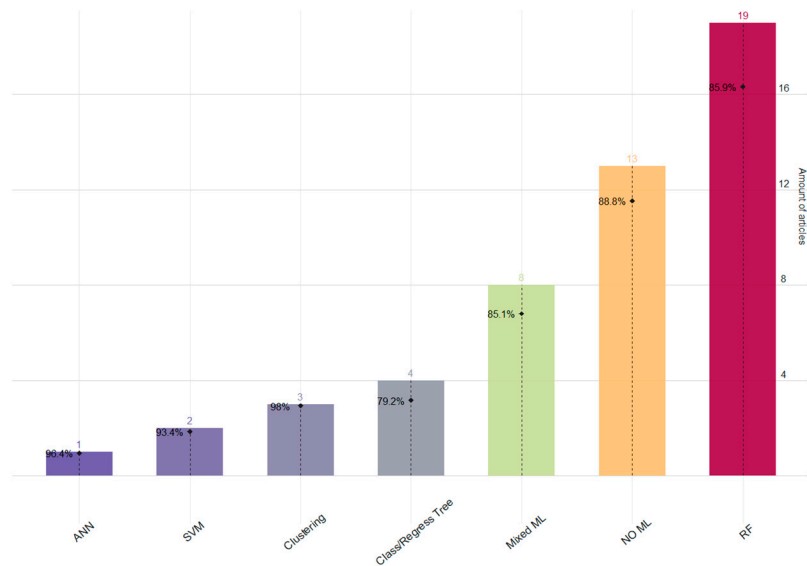

**Figure 8.** Distribution of the method used and average accuracy. The long-dash line indicates the accuracy scale from 0% to 100%; the dot marks the average accuracy (ANN = 96.4%, SVM = 93.4%, Clustering = 98%, Classification/Regression Tree = 79.2%, Mixed ML = 85.1%, NO ML = 88.8%, and RF = 85.9%).

*3.5. What Economic Gains Can Be Realized from Integrating Cloud Computing and Remote Sensing Data in the Monitoring of Wetlands?*

Cloud computing provides benefits at two levels: the first is scaling, as the user organizations save money because they purchase the cloud computing-related resources in massive quantities at lower costs, and thus can provide the services to end users at a lower cost. The second is the global reach of the companies/organizations, which also increases by using cloud computing. As a result, the end users can avoid the substantial up-front capital expenditure costs of purchasing their expensive infrastructure. As in other fields, scientists, companies, and organizations related to wetlands also benefit economically from implementing cloud computing and remote sensing data (Table 5).

Wetlands are ecosystems with very high productivity; thus, they are considered among the most economically valuable ecosystems for society [70]. Wetlands are ecosystems that offer a diverse array of ecosystem services and are regarded as vulnerable systems that exhibit rapid responses to alterations in the surrounding environment [71]. Unfortunately, in the last decades, wetlands have been lost worldwide [72], thus impacting the financial services they provide. Economic valuations of wetland services may provide a better understanding of the loss for an organization and government, but due to wetland location, cost, and time, a field survey is generally not a viable option, especially for poor or developing countries. Due to their significantly lower cost, time, and ability to monitor a large area of wetlands and their resources, cloud computing and remote sensing data may play a significant role in economic decision making by policymakers and stakeholders. Using GEE, researchers reported a significant loss in semi-arid southern African wetlands due to unsustainable use and poor management [73], thus pushing the authorities to act differently to preserve their resources. The authors also showed how cloud computing platforms might offer unique significant data handling and processing opportunities for scientists or workers with limited resources. Thus, economically favourable policies can be created for a wetland ecosystem using cloud computing and remote sensing.

**Table 5.** Cloud computing- and remote sensing-highlighted benefits and the different economic factors they influence in wetlands.

| Factors | RS—Without CC | Benefits Due to CC + RS |
|---|---|---|
| Resolution | Differs | Differs |
| Coverage | Varies | High |
| Capital expenses | High | Less |
| Cost | High | Less |
| Time | Long | Less |
| Human resources | High | Less |
| Global Reach | Limited | High |

## 4. Limitations and Potential Threats to the Validity of Cloud Computing

This review is restricted to the following parameters:

- Consideration of published data, as this study covers primary studies published until June 2022;
- Type of literature, as this SLR encompasses peer-reviewed research articles only, while conference, workshop, symposium proceedings, and grey literature, e.g., papers only published in arxiv.org, blogs, and videos, were excluded from the paper pool;
- The perspective used to show the economic benefits was not fully covered in the SLR due to the lack of accurate data to achieve this aim, but the main aim of the review was not studying and completing a meta-analysis on the economic benefits of wetlands.

Another limitation of this SLR is excluding papers not written in English. Some valuable studies may have been published in languages other than English, leading to alternative viewpoints for defining and measuring cloud-based studies using remote sensing. Finally, there are opportunities to extend our systematic review to consider the various correlates of cloud computing and its integration with remote sensing and the economic benefits.

## 5. Conclusions

The main aim of this SLR was to find out the state-of-the-art and identify the gaps in the application of remote sensing and cloud computing technologies for the monitoring of wetlands. Platform-as-a-service was the only cloud computing service model implemented in practice to monitor wetlands. The remote sensing applications for wetland monitoring implemented by CC were related to prediction, time series analysis, mapping, classification, and change detection. Among different studies, only 51% of the literature performed at a regional scale used satellite data. It should be highlighted that the up-to-date CC and RS technologies are not integrated enough to fully realize the potential of CC in wetland monitoring.

It is worth mentioning the economic benefits that could be achieved from implementing cloud computing and remote sensing for the monitoring of wetlands—the cost and time are much less than the traditional methods of wetland monitoring, apart from using different techniques and multi-data sources with less effort.

## 6. Future Work

Table 6 summarizes the primary research needs and challenges for using cloud computing and remote sensing in wetland monitoring. The identified research needs to include integrating data, tools, and programs through platform-as-a-service (PaaS) and remote sensing; the assessment of the effectiveness of combining different remote sensing data types and monitoring strategies; and the economic valuation of wetland services using cloud computing. The challenges include the high computational resources required for large datasets, the lack of evaluation of the generalization of scale adequacy, and the lack of standardization in data acquisition and processing protocols. Addressing these research

needs and challenges will improve the effectiveness and efficiency of wetland monitoring programs and enable proactive wetland management and conservation.

**Table 6.** Summary of Research Needs Identified in the SLR.

| Research Need | Description |
|---|---|
| Cloud computing adoption by integration of data, tools, and programs | Further research is needed to explore how CC and RS can integrate vast data, tools, and programs to improve global wetland monitoring and tracking. Moreover, improved efficiency, flexibility, and cost savings can be achieved using digital twins, IoT, and CC. |
| Effectiveness of combining different remote sensing data types and monitoring strategies | Research is needed to assess the benefits and limitations of different combinations of remote sensing data types and monitoring strategies. Understanding this can improve the overall effectiveness of wetland monitoring programs and enable proactive wetland management and conservation. |
| High computational resources required for large datasets | The challenges of high computational resources required for large datasets must be addressed at a national scale. |
| Evaluation of generalization of scale adequacy | Research is needed to evaluate the generalization of scale adequacy for the cloud computing methodology and its accuracy. |
| Lack of standardization and development of standardized protocols for wetland monitoring | A lack of standardization in cloud computing and remote sensing for wetland monitoring makes comparing data across different studies and regions difficult. Thus, creating standard protocols for data acquisition and processing to improve comparability and reduce errors is necessary. |
| Economic valuation of wetland services using cloud computing | Research is needed to explore the economic valuation of wetland services using cloud computing. Increased global reach, cost savings, improved resolution and coverage, less time, and lower requirements in human resources can be achieved. |

For future research, we proposed a structure of cloud computing and remote sensing technology applications integrated with the Internet of Things (IoT) to be effectively applied for monitoring wetlands to fill the gaps found (Figure 9).

Figure 9 consists of 4 layers: (1) The map shows global wetlands and potential distribution [12]. From the sites, a wetland site was zoomed in to show the proposed sensor distribution at the site level, then it was generalized to the global scale (to the other wetland sites); (2) The sensor nodes for vegetation and hydrology, the drone and the docker to be controlled, charged, and flown remotely. Tower-based gas flux measurements are proposed with the local weather station. All the sensors are linked to an IoT gateway, e.g., LoRa Wan, through a WIFI connection and then from LoRa to a private cloud. The site's shapefile crops the satellite data to save storage capacity on the cloud. The private cloud is controlled only by the researchers responsible for the specific wetland site; (3) Public cloud, which allows the data to be shared between researchers from different wetland sites. It generally pre-processes and processes collected data on the cloud and stores them in repositories; (4) The last layer; it requires resources to provide CC services such as virtual machines, servers, data storage, and security procedures. The cloud service provider oversees it. The system's backend elements help users manage all the resources required to provide the CC services. The front-end is where the user interface and analytics are seen by the user, which could be scientists, policymakers, and governments.

This system is particularly beneficial to standardize wetland measurements and is relatively easy to apply, considering that the ground base measurements and drones can validate the satellite data. Furthermore, it will help decrease the volume of software and workforce spent on measurements and data analysis, as it assures easy ways to assess data quality.

We propose the following directions to pave the way for further research:

- Development of more comprehensive remote sensing approaches at wetland sites and linking them to capture data from these heterogeneous ecosystems automatically;
- Creating public criteria for measuring and evaluating the complex ecosystem characteristics of wetlands;
- More focus on cloud computing and remote sensing, from different scenarios as proposed structure;

- Better cloud-based data sharing security and data usability for cloud analytics tools and integration with remote sensing;
- Reproducibility and open science;
- The proposal can be made in an agreement between the countries (e.g., in Supplementary File Map S2) or the research communities in the same country to agree on which collected data can be shared in the public cloud (e.g., in Supplementary File Map S1).

This SLR will be helpful for young researchers starting their careers in wetlands and wishing to apply cloud computing and remote sensing for wetland monitoring.

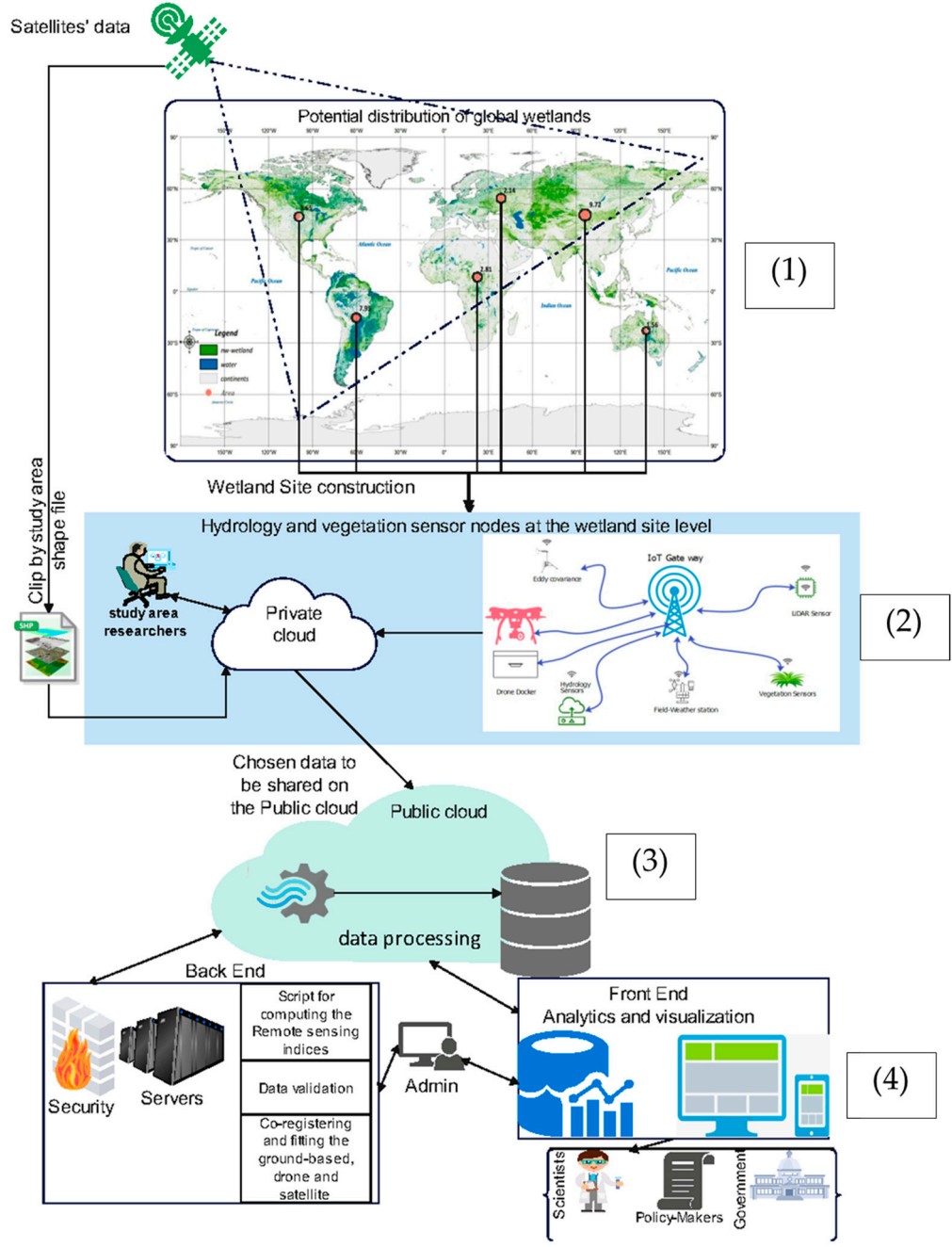

**Figure 9.** Proposed Cloud Computing structure for the monitoring of globally distributed wetland sites.

**Supplementary Materials:** The following supporting information can be downloaded at: https://www.mdpi.com/article/10.3390/rs15061660/s1, Table S1: List of the 50 articles selected from the systematic literature search for review ordered by year; Figure S1: The geographical location of the first authors of the published articles on wetland monitoring using remote sensing and cloud computing; Figure S2: The global distribution of the published articles on wetland monitoring using remote sensing and cloud computing.

**Author Contributions:** Conceptualization, data curation, methodology, formal analysis, writing—original draft preparation, A.Y.A.A.; formal analysis, data curation, writing—original draft preparation, M.A.-S.; writing—original draft preparation, review and editing, A.R.; writing—review and editing, funding acquisition, supervision, R.J. All authors have read and agreed to the published version of the manuscript.

**Funding:** National Science Centre, Poland: 2020/39/O/ST10/00775.

**Informed Consent Statement:** Not applicable.

**Data Availability Statement:** Not applicable.

**Acknowledgments:** This research was funded in whole by National Science Centre, Poland (NCN) within grant No. 2020/39/O/ST10/00775. We sincerely thank the reviewers who provided valuable comments and feedback on our systematic literature review. Their insights and suggestions were instrumental in improving the quality of our work.

**Conflicts of Interest:** The authors declare no conflict of interest.

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
