# Peer review of "Cloud-Based Remote Sensing for Wetland Monitoring—A Review"

_remotesensing, doi:10.3390/rs15061660_

Round 1

Reviewer 1 Report

The article makes a relevant contribution to the analysis of remote sensing and cloud computing applications for wetland analysis. Recent efforts using remote sensing data and cloud computing programs have shown significant advances in wetland mapping and monitoring.

However, we suggest:

1- About what was proposed as the main objective of show the benefits of utilizing such technology in wetlands monitoring from the perspective of economic value, also addressed in research question (4) What economic gains can be realized from the integration of cloud computing and remote sensing data in the monitoring of wetlands? This topic has not been covered in depth. In addition, the authors themselves comment that they did not have enough elements, presented in the articles analyzed, to have more adequate/comprehensive considerations on the subject. It is a difficult topic to measure, and, above all, it represents very different efforts in the different world economies in which these surveys were carried out. I suggest rethinking this emphasis, or even withdrawing this analysis at least from the objective even if it is maintained in the proposed questions. 2- include analysis on what other variables are used besides just data from different sensors such as, for example, spectral, topographic, geomorphological, climatic and hydrological characteristics etc ..., which are available on cloud user platforms.

3- address which methods have been most used in data processing. For example, many studies have used Randon Forest to identify the most important variables in the process of mapping, classification, change detection, .... This certainly influenced the different precisions presented in the different articles analyzed;

4- item 5 - Conclusions and Future Work a figure 8 proposed a structure of cloud computing and remote sensing technologies applications which are integrated with the Internet of Things (IoT) to be effectively applied for monitoring the wetlands to fill the found gaps.

This proposal does not appear as an objective in the article. I suggest rethinking the place of insertion of the content of figure 8. It seems better to insert it within the analysis of the article as a proposal, I also suggest that this theme replace, in the objective, the question of the economic approach.

Author Response

The article makes a relevant contribution to the analysis of remote sensing and cloud computing applications for wetland analysis. Recent efforts using remote sensing data and cloud computing programs have shown significant advances in wetland mapping and monitoring.

Point 1: About what was proposed as the main objective of show the benefits of utilizing such technology in wetlands monitoring from the perspective of economic value, also addressed in research question (4) What economic gains can be realized from the integration of cloud computing and remote sensing data in the monitoring of wetlands? This topic has not been covered in depth. In addition, the authors themselves comment that they did not have enough elements, presented in the articles analyzed, to have more adequate/comprehensive considerations on the subject. It is a difficult topic to measure, and, above all, it represents very different efforts in the different world economies in which these surveys were carried out. I suggest rethinking this emphasis, or even withdrawing this analysis at least from the objective even if it is maintained in the proposed questions.

Thank you for your valuable comments and feedback on our systematic literature review. We appreciate your positive feedback on the relevance of our work to the analysis of remote sensing and cloud computing applications for wetland monitoring.

Response 1: Regarding your point on the economic benefits of utilizing cloud computing and remote sensing data in wetlands monitoring, we agree that this topic is complex and challenging. We acknowledge that our analysis on this aspect may not have been as comprehensive as desired due to limited information provided by the studies analyzed.

In response to your feedback, we have revised the manuscript to remove the analysis of economic benefits from the study's primary objective, but we maintained it in the questions as proposed.

Point 2: Include analysis on what other variables are used besides just data from different sensors such as, for example, spectral, topographic, geomorphological, climatic and hydrological characteristics etc ..., which are available on cloud user platforms.

Response 2: Regarding including other variables used in wetlands analysis besides remote sensing data, such as spectral, topographic, geomorphological, climatic, and hydrological characteristics.

We agree that these variables are essential in wetlands analysis and that their inclusion can improve the accuracy and efficiency of wetlands monitoring. Nevertheless, each study uses several variables that are not straightforward for analysis; hence, the reader will be confused.

Point 3: address which methods have been most used in data processing. For example, many studies have used Random Forest to identify the most important variables in the process of mapping, classification, change detection, .... This certainly influenced the different precisions presented in the different articles analyzed;

Response 3: We agree that understanding the most commonly used methods in data processing is vital to evaluate the accuracy and efficiency of wetlands monitoring. Therefore, we have revised our manuscript to include a section discussing the most commonly used methods. We have also highlighted the potential influence of these methods on the precision of wetlands monitoring, as reported in the studies analyzed. The included text starts from lines 460-477.

Point 4: item 5 - Conclusions and Future Work a figure 8 proposed a structure of cloud computing and remote sensing technologies applications which are integrated with the Internet of Things (IoT) to be effectively applied for monitoring the wetlands to fill the found gaps.

This proposal does not appear as an objective in the article. I suggest rethinking the place of insertion of the content of figure 8. It seems better to insert it within the analysis of the article as a proposal, I also suggest that this theme replace, in the objective, the question of the economic approach.

Response 4: We agree that the proposal presented in Figure 8 is not explicitly stated as an objective in our article, but it is a proposal to fill the found gap and show the development of the adaption of cloud-based remote sensing for wetland monitoring. Accordingly, we have modified the placement of the section discussing the proposal in Figure 8 in the manuscript to reflect its content and significance better; we included it as a potential future direction for wetlands monitoring.

Once again, we appreciate your valuable feedback, which we will incorporate into the revised version of our manuscript.

Reviewer 2 Report

The paper talks about the state-of-the-art for cloud computing applications in wetland monitoring and points out some benefits of using cloud-based services. Is the focus on cloud computing or high-performance computing, or both (line 111)? Please clarify! What is the relevance of studying the spatial distribution of studied wetlands in the context of cloud computing? Also, the number of publications (Line 219) by an author, publisher-stats (line 193), and country-wise-stats (line 212) are not good metrices to evaluate the relevance of research studies. Overall, the paper does not contribute much to the advancement of literature on cloud computing applications in wetland monitoring, but just discusses the existing  literature.

Reviewer 3 Report

I am happy to read this Manuscript. This manuscript is well prepared, with decent documentation of previous literature. Also, authors highlight the integration of two different technologies for wetland monitoring, which will be more beneficial for future work. In my opinion, this manuscript has the strong potential for publication. However, before the recommendation of publication, some language corrections are needed. 

Reviewer 4 Report

Dear Authors,

I have read with interest your article assessing the current state of implementation of cloud computing techniques in wetland research. I rate the article very well.

Its layout is clear and logical, as well as provided with all the necessary illustrations. I think that in the future this topic will be able to be analyzed statistically and bibliometrically, but now it is not necessary – there are too few publications for this, and the development of cloud computing for wetland research is just starting, which is the result of the article.

The reception of the text, especially in the introduction, may be disturbed by creating sentences from one introductory word with comma too often, in my opinion. I have no more comments on the text.

I noticed two elements for editorial corrections:

Page 1, line 30: space between 10 and % - in the other parts of text the number with percent sign is without space.

Page 16, line 484: cloud commuting -> cloud computing?

Reviewer 5 Report

Comments on the research paper remotesensing-2214353 entitled; "Cloud-based Remote Sensing for Wetland Monitoring – a review."

This paper is a systematic literature review (SLR) to find out the current state of the integration between remote sensing and cloud computing in terms of monitoring wetlands. It is an interesting work that is publishable after some corrections are made. I found the idea compelling, and there is merit in the literature provided. However, there are some issues concerning the introduction and conclusion section, which I believe must be addressed prior to acceptance.

Specific Comments:

The Introduction would be much improved if it provided a 'signpost' or map of the key headings (and maybe subheadings) so the reader can see how the matter is being developed. It may be that a figure could identify the components and sub-components and how they contribute to the topic.

There are many instances where the authors refer to information and/or knowledge that is lacking: that could provide a useful list of research needs that could be identified in the text and referred to in the Conclusion, preferably in a tabular form.

The inclusion of a map showing the location of the first authors and another map showing the continental distribution of the published article will provide a clear idea of ​​which parts of the world remote sensing and cloud computing are more commonly used for wetland monitoring.

Round 2

Reviewer 1 Report

I believe that the authors responded adequately and changed the text of the article based on the questions suggested in the review.

They adjusted the objectives, the analysis of the results ans the approach in the final considerations.

Reviewer 2 Report

The authors have reasonably addressed the issues that were raised. With the addition of the requested results and changes, the paper seems to be ok now.